# Ginseng Soluble Dietary Fiber Reverses Obesity via the PPAR/AMPK Signaling Pathway and Improves Intestinal Flora in Mice

**DOI:** 10.3390/foods14101716

**Published:** 2025-05-12

**Authors:** Yue Zhang, Chen Bai, Jiyue Sha, Xiaohui Huo, Di Qu, Jianbo Chen

**Affiliations:** Institute of Special Animals and Plants, Chinese Academy of Agricultural Sciences, Changchun 130112, China; zy135178@163.com (Y.Z.); baichen8687@163.com (C.B.); s18946576631@163.com (J.S.); xiaohui.114@163.com (X.H.); qudi_caas@163.com (D.Q.)

**Keywords:** ginseng soluble dietary fiber, PPAR/AMPK, metabolism, gut microbiota, obesity

## Abstract

Background: Ginseng soluble dietary fiber (GSDF) has been shown to have good physicochemical properties; however, its in vivo benefits in obesity are yet to be fully elucidated. Methods: To explore this, C57BL/6J obese mice were given metformin hydrochloride and different doses of GSDF for 60 days. The levels of blood lipids and inflammatory factors were detected by ELISA, and the pathological alterations were detected through the application of HE staining. The level of adipose tissue protein in epididymis was detected by Western blotting and through the effects of 16S rRNA sequencing on gut microbiota. Results: The results showed that GSDF significantly improved basal physiological indices, lipid levels, and serum cytokine levels in the obese mice. GSDF increased the expression levels of PPAR-γ, AMPK, and P-AMPK proteins, and lowered the expression of IL-1β, TNF-α, and other proteins in the adipose tissues of the epididymis, in turn inhibiting adipogenesis and ameliorating lipid metabolism disorders. By lowering the *Firmicutes/Bacteroidetes* ratio in the gut and altering the abundance of thick-walled bacteria and mycobacterium, the abundance of species such as *Lactobacillus*, *Alloprevotella*, and *Faecalibaculum* was altered to improve cecum health. Conclusions: These results suggest that GSDF may have a positive effect on growth, obesity, and cecal health in obese mice.

## 1. Introduction

With the continuous improvement in quality of life, the dietary habits of people globally tend to be high in sugar and fat, resulting in an increase in the number of obese people over time. Obesity is a chronic disease caused by abnormal lipid metabolism and its incidence is increasing [1]. It encompasses not only an increase in body weight but also the abnormal build-up of adipose tissue within the body [2]. When the nutrient storage system within the body is subjected to an extended duration of excessive energy, its energy storage capacity is greater than the effective use of energy, eventually leading to obesity. Obesity has thus been shown to be closely related to many chronic diseases [3], including nonalcoholic fatty liver disease [4], hyperlipidemia [5], diabetes [6], and atherosclerosis [7]. Numerous research investigations have revealed a strong correlation between obesity and disturbances in intestinal health. The human gut microbiota is abundant and maintains a dynamic balance between symbiosis and competition with the host. Under normal physiological conditions, beneficial bacteria synthesize various proteins and vitamins to meet the normal needs of the human body. In contrast, abnormal or local transfer of the number and activity of microorganisms may lead to dysbiosis of the microbiota, increasing the number of harmful bacteria, thus producing direct or indirect pathogenic effects on the host [8]. The gut microbiota of people with obesity is often abnormal; therefore, nutrients that cannot be absorbed by the body are reabsorbed and accumulate in cells, eventually leading to various symptoms such as weight gain, hyperlipidemia, and hyperglycemia [9].

In recent years, many natural active ingredients have been extracted from plants and their role in improving obesity and intestinal health has attracted increasing attention. Dietary fiber (DF) is an important component that can be divided into water-soluble dietary fiber (SDF) and non-water-soluble dietary fiber (IDF), according to its aqueous solubility. DF is mainly derived from plant cell walls and exhibits strong water absorption, which can enhance the sense of satiety and reduce food and caloric intake [10]. In addition, it can positively affect weight management and metabolic health by regulating physiological mechanisms such as fat storage and glucose metabolism [11]. Many studies have shown that DF can affect energy metabolism and fat storage by regulating the AMPK pathway, inflammatory factors, and other proteins [12,13,14]. The AMPK signaling pathway forms a heterotrimeric structure, with its central role being to react to fluctuations in the intracellular concentrations of ATP and AMP. When intracellular ATP levels decrease and AMP levels increase, AMPK is activated, prompting the cellular metabolic mode to shift from anabolic to catabolic, thereby enhancing energy supply [15]. The AMPK signaling pathway plays a key role in various metabolic regulation processes such as cell growth, differentiation, and inflammatory responses [16]. In addition, peroxisome proliferator activated receptor gamma (PPAR-γ) stands as a crucial constituent within the PPAR family, and is one of the main regulators of metabolic changes caused by an obese diet. The expression level of PPAR-γ in the skeletal muscle of patients with obesity generally shows a downward trend [17]. This change in expression is not only related to obesity, but also affects the growth arrest process of precursor adipocytes before differentiation [18].

Ginseng is a medicinal and edible plant that is rich in nutrients, safe, non-toxic, and has high medicinal and nutritional value [19]. Its excellent anticancer [20], anti-aging [21], sleep improvement [22], and immune regulatory functions [23] help maintain immune balance and health. Ginseng dietary fiber (GDF) is obtained following the enzymatic hydrolysis and purification of ginseng residue after the extraction of active ingredients. It is rich in sugars, proteins, and essential amino acids for the human body, which account for 70% of ginseng residue; ginseng water-soluble dietary fiber (GSDF) accounts for approximately 10% and ginseng non-water-soluble dietary fiber (GIDF) accounts for approximately 60% [24]. Our previous study showed that GSDF is an acidic heteropolysaccharide (uric acid content of 4.42%) rich in protein, amino acids, and mineral elements; it is not only rich in nutrients, but also has an easily adsorbed microstructure, a large amount of antioxidant functional groups, and ideal processing properties, which can effectively promote the absorption of nutrients by the human body. Because of its easy accessibility, good aqueous solubility, richness in nutrients and oil holding capacity, and “low heat”, GSDF delays glucose diffusion and inhibits α-amylase/α-glucosidase. GSDF exhibits extensive application prospects within the realms of nutrition and medicine [25].

However, to date, there are no reports on the effects of GSDF on obesity. Although GSDF has attracted much attention owing to its unique physicochemical properties and positive effects on the gut microbiota, its specific mechanism of action in alleviating hyperlipidemia (i.e., obesity-related hyperlipidemia) remains unclear. The objective of this research was to investigate the impact of GSDF on reducing lipid levels and enhancing intestinal health in obese mice, providing experimental data to support its use in improving high-fat diet (HFD)-induced obesity. These findings are meant to promote diversification of the ginseng industry and provide a reference for the development of other plant fibers.

## 2. Materials and Methods

### 2.1. Materials

Heat-resistant α-amylase, glacial acetic acid (acetic acid), alkaline protease, and starch glucosidase solutions were obtained from Santa Cruz Biotechnology (Dallas, TX, USA). This study used 5-year-old ginseng (Jilin Zhongsheng Pharmaceutical Co., Ltd., Baishan, China). Triglyceride (TG), total cholesterol (TC), low-density lipoprotein cholesterol (LDL-C), and high-density lipoprotein cholesterol (HDL-C) kits were obtained from Nanjing Jiancheng Bioengineering Research Institute. PPAY-γ (16643-1-AP), AMPK (10929-2-AP), P-AMPK α (2535T), IL-1β (ab234437), TNF-α (17590-1-AP), and β-actin (# T0022) were purchased from Changchun Yuanda Biotechnology Co., Ltd. (Changchun, China).

### 2.2. Preparation of GSDF

The extraction method of GSDF used in this study is as described by Zhang et al. (2024) [26]. Briefly, ginseng was boiled twice in water (1:10), each time for 2 h, and dried at 60 °C. The samples were ground using a FW135 crusher (Tess Instrument Co., Ltd., Tianjin, China). After crushing, the residue was washed twice with 85% ethanol and distilled water to remove impurities, and then dried again at 60 °C for 24 h. The remaining residue was filtered through a 60-mesh sieve and then placed in sterile bags, being stored at −20 °C pending subsequent analysis. The entire production process ensured the quality and purity of the ginseng residue, and provided reliable raw materials for subsequent experiments. We prepared 1.2 L of a sodium bicarbonate solution with a solid–liquid ratio of 1:15 and adjusted the pH to 8.2. Subsequently, 1% heat-stable α-amylase and 1% alkaline protease were added to the mixture. Ginseng residue (10,800 g) was mixed at 300 rpm for 120 min at 60 °C to extract GSDF. Subsequently, the pH of the mixture was adjusted to 4.5 with acetic acid and 3% starch glucosidase was added. Under the same conditions, mixing continued for 120 min. Following this, the mixture underwent centrifugation at a force of 8000× *g* for a duration of 15 min, following which the solid residue was retrieved.

### 2.3. Animal Experimental Design

The entirety of animal experiments conducted received approval from the Animal Management and Ethics Committee of the Institute of Zoology and Botany, Chinese Academy of Agricultural Sciences, as per the stipulated license number ISAPSAEC-2022-66. The feeding conditions were strictly set in accordance with the national standard “Laboratory Animal Environment and Facilities” (GB14925-2010) [26]. The experimental animals were SPF-grade male C57BL/6J mice (6 weeks old, 18.03 ± 0.5 g, identification number: SCXK (liao) 2020-0001), and the animal feed was procured from Liaoning Changsheng Biotechnology Co., Ltd., (Benxi, China). Drinking water was subjected to high-pressure sterilization. Forty-eight mice were housed in SPF animal rooms (temperature 20–22 °C, humidity 50–55%) with lighting conditions set to 12 h during the day and 12 h at night. A standard diet containing 24% protein, 13% fat, and 63% carbohydrates and a high-fat diet of 60 KJ fat calories were used. Systematic replacement of the drinking water and the bedding was implemented on a regular basis. After 7 days of adaptation, the mice were randomly divided into six groups (*n* = 8): the high-fat-diet group (HFD group), control group (physiological saline, control group), low-dose GSDF (150 mg/kg·bw, GSDF-L group), medium-dose group (300 mg/kg·bw, GSDF-M group), high-dose group (600 mg/kg·bw, GSDF-H group), and metformin hydrochloride-positive control group (200 mg/kg·bw, MH group). The control group was given 0.2 mL of physiological saline and standard feed every morning, whereas the intervention group was given high-fat feed and different doses of GSDF or MH (dissolved in physiological saline) 0.1 mL/10 g·bw, for a total of 60 days.

### 2.4. Determination of Basic Indicators

Upon completion of the experimental period, the mice were subjected to a fasting phase lasting 10 to 12 h. The following day, the body weights and fasting blood glucose concentrations of the mice were documented. Mice were euthanized with excess carbon dioxide and immediately after collection, spleen, kidney, liver, and brown adipose tissue (BAT) were dissected and weighed for calculation of organ index. Organ index = organ wet mass (mg)/mouse body weight (g). The kidneys, liver, spleen, and BAT were stored in a freezer at −80 °C.

### 2.5. Serum Biochemical Index and ELISA Assay

The mice were euthanized using excessive carbon dioxide. Blood was collected and spun in a centrifuge at 3000 rpm for a duration of 15 min while maintained at 4 °C in order to quantify the serum concentrations of TC, TG, LDL-C, and HDL-C (Nanjing Jianjian Bioengineering Institute, Nanjing, China). The levels of cytokines IL-1β and IL-10 in the mice serum were assessed by strictly adhering to the guidelines provided by the manufacturer (Shanghai Enzyme Linked Biotechnology Co., Ltd., Shanghai, China).

### 2.6. Organizational Analysis

Mice were dissected to collect white adipose tissue from the epididymis (eWAT) and it was weighed immediately after collection. White adipose tissue was separated from the mesentery (mWAT) and colon tissue, and the length of the colon was measured, weighed immediately after collection, and processed for further analyses. Colonic tissue and eWAT were embedded in 10% neutral formalin and embedded with Parafn. Each of the tissue samples was cut into 4-micrometer-thick sections and subsequently stained using hematoxylin and eosin (H&E, E607318-0200, BBI). All images were observed and captured at the same intensity using a microscope (BX53; Olympus, Tokyo, Japan).

### 2.7. Western Blotting

Protein concentration was quantified by the BCA method, and then an aliquot of 40 μg of total protein from each sample was separated by SDS-PAGE separation gel, transferred onto a PVDF membrane, and subsequently subjected to blocking treatment using skimmed milk or 5% BSA for a duration of 1 h, followed by incubation with primary antibodies, including PPAY-γ, AMPK, P-AMPKα, IL-1β, and TNF-α overnight at 4 °C. Following this, incubation was performed with species-specific secondary antibodies for 1 h. ECLA and reagent B were mixed in equal volumes, and the membrane made of PVDF was in complete contact with the developer and exposed to the universal imaging system. The Western blot band was analyzed using Image G image analysis. Image G image analysis was further used to determine the net grayscale value, which was compared with the internal reference β-actin, the ratio was calculated, and the differences between groups were compared.

### 2.8. Microbiome Analysis

Following the final dose administration, every mouse was subjected to a 3 h fasting period, and sterile mice feces (for 16S rRNA sequencing) were collected and stored in a freezer at −80 °C. Approximately 0.30 g of feces from each mouse was stored on dry ice and sent to Beijing Novogene Biotechnology Co., Ltd. (Beijing, China) for gut microbiota 16S rRNA testing. Microbial DNA was extracted from samples using a fecal DNA extraction kit. DNA was quantified using an ultra-micro spectrophotometer, and its integrity, purity, fragment size, and concentrations were determined through agarose gel electrophoresis. For primer design, the V3–V4 segment of the 16S rRNA gene (a) was chosen as the target for polymerase chain reaction amplification (forward primer: reverse primer; reverse primer).

### 2.9. Statistical Analysis

The findings are reported as the mean value ± SEM. All statistical analyses were conducted utilizing GraphPad Prism 10.0 software. To ascertain differences among the groups, a one-way analysis of variance was employed, with statistical significance established at *p* < 0.05.

## 3. Results

### 3.1. Effect of GSDF on Basic Indicators in HFD-Induced Obese Mice

In this study, from the 19th day, the weight of the mice increased significantly, and after the administration of GSDF, the weight gain decreased significantly. On the 60th day, the body size of the HFD group was significantly larger than that of the other groups (*p* < 0.05), the hair was sparse and greasy, and the weight gain rate and fasting blood glucose level were significantly increased. Compared to the HFD group, the weight, weight gain rate, and mean fasting blood glucose of the GSDF and MH groups were significantly reduced (Figure 1a–c), and those for the MH-, low-, medium-, and high-dose groups were significantly improved. The relative organ indices of both the liver and kidneys exhibited a notable and statistically significant elevation in the HFD group. Compared to the HFD group, the GSDF and MH treatment groups showed significantly reduced liver organ index levels, and the GSDF-L, GSDF-M and MH treatment groups showed significantly reduced kidney organ index levels but GSDF-H was not significant. The spleen organ index did not show any notable variation (Figure 1d–f). A marked reduction was noted in the proportion of BAT within the HFD group, whereas that in the GSDF-L, GSDF-M, and MH groups was significantly increased, but GSDF-H was not significant. Among them, the proportion of BAT in the GSDF-M group returned to that of the control group (Figure 1g). Therefore, GSDF can improve the body size and fur condition of obese mice and reduce their weight, fasting blood glucose levels, and organ indexes, thereby normalizing lipid levels.

### 3.2. Effect of GSDF on Blood Lipids and Inflammatory Factors

In the HFD group of mice, a notable and significant elevation was observed in the serum concentrations of TG, TC, LDL-C, and HDL-C (Figure 2a–d). Compared with the HFD group, the levels of TG and TC in the serum of GSDF- and MH-group mice were significantly decreased, and the improvement effects of the GSDF-L, GSDF-M, and GSDF-H groups were significant. The LDL-C levels of mice in the GSDF-M and MH groups decreased significantly, with that of the GSDF-M group showing the best improvement. Compared with the HFD group, the HDL-C levels of the GSDF-M- and MH-group mice were significantly decreased and returned to the level of the control group, but there was no statistically meaningful improvement in the GSDF-L and GSDF-H groups. In the HFD group, the content of IL-10 in serum was decreased, while that of IL-1β was significantly increased. Compared with the HFD group, the level of IL-10 in the GSDF-M group was significantly increased, but the improvement in the GSDF-L and GSDF-H groups was not significant. In contrast to the HFD group, the content of IL-1β in the GSDF-M and MH groups was significantly reduced. Contrarily, the improvement effect in the GSDF-L group was not significant and no improvement at all was evident in the GSDF-H group (Figure 2e,f). GSDF improves blood lipid levels of obese mice and effectively regulates cytokine levels in the serum of these obese mice.

### 3.3. Effects of GSDF on Histopathology in HFD-Induced Obese Mice

In the HFD group, the eWAT index was significantly increased, eWAT was enlarged, cells were disordered and loose, and vacuoles were increased. In contrast to the HFD group, the GSDF and MH groups exhibited a considerable and significant reduction in their eWAT indices, epididymal adipocytes were arranged in an orderly manner, and the number of vacuoles was reduced. Therefore, GSDF improved and restored the proportion of eWAT in obese mice (Figure 3a,d). In the HFD group, the mWAT index was significantly increased, and colon length was significantly shortened.

Through H&E staining, it was revealed that the architectural integrity of the colon in mice belonging to the control group remained uncompromised, villi on the surface of the mucosal epithelium were arranged in an orderly manner, and goblet cells were abundant. Within the HFD group, the colonic villi presented with a reduced length and a sparse distribution pattern. The epithelial cells were arranged in a disorganized fashion, prominent interstitial edema was discernible, and the crypts displayed a shallow morphology. Compared to the HFD group, the colon length of obese mice after GSDF intervention was significantly increased, the villi were orderly and dense, the crypts were deepened, villus interstitial edema was reduced, and the number of goblet cells was increased. The recovery of colonic tissue in the MH, GSDF-M, and GSDF-H groups was less pronounced than that in the GSDF-L group (Figure 3b,c,e). GSDF ameliorates the damage caused by an HFD to the intestinal tissue of mice and restores the integrity of the intestinal structure of mice to a certain extent.

### 3.4. Effect of GSDF on the PPAR/AMPK Pathway

In an effort to unravel the molecular mechanism through which GSDF exerts its regulatory influence on lipid metabolism and inflammation-associated proteins, we conducted an assessment of the protein expression levels within the AMPK/PPAR signaling pathway. The results showed that the protein expression levels of PPAR-γ, AMPK, and P-AMPK were significantly decreased in the HFD group. Compared with the HFD group, the GSDF and MH groups significantly increased the protein expressions of PPAR-γ and AMPK. The MH, GSDF-L, and GSDF-M groups also showed significantly increased P-AMPK expression (Figure 4a,d–f). GSDF inhibits the synthesis of TG and TC, reduces the increase in fat, and improves lipid metabolism by activating the intracellular PPAR/AMPK signaling pathway in mice with obesity.

The protein expression levels of IL-1β and TNF-α were significantly increased in the HFD group. Compared with the HFD group, the MH and GSDF groups showed decreased expression of inflammatory proteins, in which the expression of IL-1β and TNF-α was significantly decreased in the MH, GSDF-L, and GSDF-M groups, and that of IL-1β was significantly decreased in the GSDF-H group (Figure 4b,c). GSDF improves tissue inflammation in mice caused by lipid metabolism disorders by inhibiting the expression of IL-1β, TNF-α, and other cell proteins. It can effectively reduce blood lipids by activating the PPAR/AMPK pathway, inhibiting lipogenesis, promoting lipolysis, and regulating the protein expression level of key lipid synthesis enzymes. GSDF induces a decrease in the expression of inflammatory proteins like IL-1β and TNF-α in tissues through its regulatory action, inhibits the occurrence of tissue inflammation, and improves the damage to eWAT caused by obesity.

### 3.5. Composition and Diversity of Mouse Intestinal Microflora

The effect of GSDF on the composition of gut microbiota in mice administered an HFD was investigated. The microbial communities of the cecal samples were analyzed using 16S rRNA gene sequencing. Based on the preliminary experimental results, a comparative assessment was carried out to examine the disparities among the control group, the HFD group, and various GSDF groups. A total of 2,294,973 high-quality sequences were obtained from 36 samples in the six groups, with a total length of 947,222,541 bp. This particular system was employed to investigate the impact of varying dosages of GSDF on the gut microbiota architecture of obese mice (Figure 5a). The findings suggest that, given the current sequencing depth, the sequencing data obtained from the samples are adequate to showcase the inherent diversity within them. Consequently, each sample was deemed suitable for subsequent analytical procedures. From Figure 5b, an analysis based on the Chao-1 index for estimating microbial richness revealed that the control group exhibited a significantly greater level of richness compared to the HFD, MH, and GSDF groups (*p* < 0.05), and the richness of the GSDF-H group was higher than that of the GSDF-M group (*p* < 0.05). There was no statistically significant difference in the Chao-1 index between the HFD and GSDF groups. Shannon and Simpson indices were used to analyze microbial abundance. The Simpson and Shannon indices of the HFD group were significantly lower than those of the control and GSDF-L groups (*p* < 0.05), whereas the Shannon index of the GSDF-L group was significantly higher than that of the MH group (*p* < 0.05). These alpha diversity results indicate that an HFD reduces the richness and diversity of gut microbiota in mice and that GSDF intervention can alleviate this reduction to some extent. The β diversity index was used to evaluate the differences in microbial community composition between samples, using weighted and unweighted UniFrac distances. Principal coordinate analysis (PCA) indicated that samples belonging to the same group have similar relationships. Each group of samples was also relatively clustered together without obvious partitioning. PCoA based on the unweighted UniFrac distance revealed no clear division between samples within the group. As shown in Figure 5c, the PCoA indicated a significant difference in the number of points between the HFD group and the representative sample of the control group, indicating a significant variance in the gut microbiota structure of the mice after modeling. Most points in the MH group and different GSDF group samples clustered together, indicating a similar microbial community composition and structure. In summary, the microbial community structures within the groups were highly similar. If species abundance is not considered, differences in gut microbiota occur after long-term intake of an HFD, with GSDF intervention not having a significant effect on β diversity.

### 3.6. Analysis of GSDF in Gut Microbiota Composition

After 16S rRNA sequencing, we examined the structure of the gut microbiota at the phylum, genus, and other taxonomic levels. At the phylum level, the dominant bacterial species in the gut microbiota were *Firmicutes*, *Bacteroidetes*, *Actinobacteria*, *Proteobacteria*, and *Verruca*. *Bacteroidetes* and *Firmicutes* were the most common phyla among all microbial categories (Figure 6b). The relative richness of *Firmicutes* increased in the HFD group, whereas those of *Bacteroidetes*, *Proteobacteria*, and *Verrucobacteria* decreased. At the phylum level, compared with group C (average = 6.922), the HFD group (average = 90.233) showed a significant increase in the proportion of *Firmicutes* and *Bacteroidetes* (*p* < 0.01), whereas GSDF-L (average = 20.197) intervention showed a significant decrease (*p* < 0.01). Compared to the GSDF-L group, both the MH and GSDF-M groups showed significant increases (*p* < 0.05). Compared to group C, the HFD group showed an increasing trend in the numbers of *Firmicutes*, *Deinococcus*, *Campylobacter*, *Desulfobacteria*, and *Patescibacteria*, whereas the numbers of *Bacteroidetes*, *Actinobacteria*, and *Proteobacteria* decreased significantly. Compared to the HFD group, the abundance of *Campylobacter* and *Patescibacter* in the gut microbiota of mice fed GSDF was significantly reduced (Figure 6a). Heat map analysis at the genus level showed that, compared with group C, the HFD group had an abundance in harmful bacteria, such as *Candidatus_Saccharimonas*, *Romboutsia[Ruminococcus] torques_group*, *Mycoplasma*, *UBA1819*, *Helicobacter*, *Mucispirillum*, *Anaerotruncus*, *Dubosiella*, and *Erysipelatocolstridium*. The relative abundance of beneficial bacteria such as lactic acid bacteria and *Bacteroidetes* was reduced, but the difference was not significant (Figure 7). Comparatively to the HFD group, the gut microbiota of the different GSDF groups showed some recovery. The comparative proportion of bacteria with beneficial effects, such as *Bacteroides*, unidentified *Oscillospiraceae*, and *Blautia*, increased in the GSDF-L group. The GSDF-M group showed beneficial bacteria, such as *Halomonas*, *Gordonibacter*, and *Ligulatobacilli*. The GSDF-H group exhibited a heightened relative proportion of bacteria with beneficial physiological effects such as *Coriobacteraceae UCG-002*, *Lactobacillus*, and *AlloPrevotella*. Different doses of GSDF increased the relative abundance of *lactobacilli* and *Bacteroidetes*, decreased the relative abundance of *Escherichia coli*, and affected the composition of the gut microbiota. According to the column chart analysis of species abundance at the subordinate level, compared to group C, the HFD group showed a significant increase in the genus contents of *Dubosiella*, *Colidextribacter*, *Mycoplasma*, and *Helicobacter*, whereas the genera *Lactobacillus* and *Lachnospiraceae NK4A136_group*, *Faecalibaculum*, and *AlloPrevotella* decreased significantly. Compared with the HFD group, the relative abundance of *Dubosiella*, *Helicobacter*, and *Mycoplasma* in the GSDF group was significantly reduced. In addition, after GSDF intervention, the relative abundance of *Faecalibaculum*, *Colidextribacter*, *Lachnospiraceae NK4A136_group*, *GCA-900066575*, *Lactobacillus*, and *AlloPrevotella* increased.

### 3.7. LEfSe Analysis

Linear discriminant analysis effect size (LEfSe) was employed to assess the statistical significance of the disparities in abundance observed among microbial communities. We conducted LEfSe analysis on biomarkers with LDA scores greater than the set value (default set to 4), indicating statistical differences between groups, to further explore all biomarkers with statistical differences between groups. There were statistical differences among the 22 microbial communities in the three groups (Figure 8), among which the abundance of *Bacteroidales Bacteroidia*, *Bacteroiota*, *Muribaculaceae*, *Faecalibaculum*, *Lachnospiraceae NK4A136_group*, *Oscillospiraceae*, and *Tirchinella_pseudospiralis* had a significant impact on group C. The main strains in the HFD group were *Dubosiella*, *Helicobacter_ganmani*, *Ruminococcus torques_group*, *Peptostreptococcae*, and *Romboutsia*. The dominant bacterial strains in the GSDF intervention group were *Lachnospirales*, *Lachnospirales*, *Desulfovibrio ceraceae*, *Desulfovibrio cerales*, *Desulfobacter ota*, and *Lachnospiraceae bacterium DW59*. These results indicate that GSDF intervention can partially reshape the microbial community structure in mice.

## 4. Discussion

Intestinal health largely depends on diet, and gastrointestinal inflammation induced by a long-term HFD is considered a potential cause of obesity-induced intestinal complications [27]. A long-term high-frequency fat diet can lead to obesity, aggravate colitis [28], and induce colon cancer [29]. BAT has a unique structure and function; its core function is to generate heat [30]. When the human body needs to increase heat, uncoupling protein 1 (UCP1) in the mitochondria of BAT cells causes brown adipocytes to produce heat during energy consumption. This process reduces fat accumulation in the body by promoting the oxidative decomposition of fatty acids, in turn improving the basal metabolic rate and reducing the risk of obesity [31]. In obese individuals, BAT content and activity may be reduced [32]. Our results showed that GSDF effectively enhanced the BAT index and improved HFD-induced obesity.

To clarify the mechanism of GSDF effects on adipose tissue caused by obesity, we examined the PPAR/AMPK pathway and inflammatory proteins in eWAT. As a transcription factor that regulates lipogenesis, the AMPK-mediated signaling pathway serves a crucial function in orchestrating cellular metabolic activities [33]. During the recent period, the number of studies on the regulation of lipid metabolism by AMPK has increased. One research investigation demonstrated that the soluble dietary fibers derived from lotus root modulate adipose tissue expansion in obese mice. This modulation occurs through the regulation of the AMPK signaling cascade and the suppression of the activity of the transcription factor SREBP-1, along with the lipogenic genes under its regulatory control [34]. Soluble dietary fiber from wheat bran regulates trimethylamine metabolism and co-regulates the host AMPK pathway by modulating the intestinal flora, ultimately reducing obesity [12]. Flaxseed polysaccharide, by activating the AMPK signaling pathway, accelerates the process of lipid metabolism, eliminating leptin resistance and improving anti-obesity efficacy. This suggests that flaxseed polysaccharide can become an activator of the AMPK signaling pathway [35]. In this study, GSDF was found to activate AMPK due to its polysaccharide structure, which not only inhibited adipogenesis, but also improved glucose utilization and lowered blood glucose levels. In addition, the activation of AMPK affects the composition and metabolic activity of the intestinal flora and finely regulates fat metabolism and energy balance [36]. PPAR-γ plays a central regulatory role in metabolic changes induced by an obese diet. It can activate a variety of genes directly involved in lipid storage or release and adipocyte differentiation. A recent study showed that konjac glucomannan, as a new dietary fiber supplement, promotes metabolism through PPAR-γ protein and regulates intestinal flora to enhance obesity caused by high-fat blood [37]. In the case of adipose tissue expansion accompanied by functional impairment, GSDF promotes the transformation of white adipocytes to brown adipocytes by activating the PPAR-γ signaling pathway. Simultaneously, it also reduces the levels of a variety of inflammatory factors [38], thereby playing a protective role in damaged tissues [39]. The specific mechanisms of the PPAR-γ and AMPK signaling pathways in regulating obesity are different, but there are interactions between them, which jointly affect the development and progression of obesity [40]. For example, some studies have shown that PPAR-γ activators such as rosiglitazone and pioglitazone can increase adipocyte sensitivity to insulin and improve glucose metabolism by activating PPAR-γ, and this process may involve the activation of AMPK [41]. The activation of AMPK may also affect the activity and function of PPAR-γ, thereby further regulating fat metabolism and energy balance. In obesity, adipocytes produce and release a variety of inflammatory factors, such as TNF- α and IL-1β. The excessive release of these factors leads to a low-grade chronic inflammatory response. With an increase in inflammatory factors in adipose tissue, the surrounding immune cells are activated and release inflammatory factors, further exacerbating the inflammatory response [42]. Therefore, GSDF has great potential for the treatment of hyperlipidemia and fat differentiation by regulating PPAR/AMPK and other signaling pathways and inflammatory factor protein content. The present study suggests that the differential performances presented by the three different-dose GSDF groups are most likely closely linked to their respective structural properties. Polysaccharides, as the key components of GSDF, have a role in the biological activity of polysaccharides in terms of their molecular weight, the proportion of various monosaccharides, the specific type of glycosidic bonds, and the overall spatial conformation. In view of this, the polysaccharides in the three different doses of GSDF groups may have significant differences in the above structural features, which is the reason why different doses of GSDF have different effects in improving obesity and regulating intestinal flora. The intestinal flora can produce harmful metabolites, which are closely related to the disturbance of intestinal barrier function and the induction of disease. In contrast, it is capable of generating advantageous metabolites that possess anti-inflammatory properties and functions in safeguarding the intestinal barrier. These metabolites are expected to exert a far-reaching influence on the host’s health status and the progression of diseases. Therefore, the structural balance of the intestinal flora plays a vital role in maintaining body health. Among the many influencing factors, dietary habits are recognized as key factors determining the composition of intestinal flora [43]. Herein, our findings indicate a significant rise in the abundance of *Firmicutes* in the cecal flora of obese mice, coupled with a considerable reduction in the levels of beneficial bacteria, notably *Bacteroidetes*. This led to an overall increase in the *Firmicutes*-to-*Bacteroidetes* (F/B) ratio. Among the beneficial bacteria, *Firmicutes* are a key constituent of the human intestinal microbiome, with a primary function in energy absorption from the diet [44]. Studies have shown that in overweight men, the content of *Firmicutes* in their intestines increases after eating foods rich in high-resistance starch [45]. In animal models of obesity and obese individuals, the F/B ratio in the intestinal flora is generally increased [46]. The findings of this result imply that, at the phylum level, an elevation in the quantity of *Firmicutes* could represent one of the underlying mechanisms through which GSDF facilitates a more efficient energy-harvesting process from the diet by the host and expedites the generation of metabolites. The influence exerted by MH and *Firmicutes* content in the cecal flora in this study was similar to that of GSDF. Simultaneously, the beneficial *Bacteroidetes* of the intestinal flora increased significantly after GSDF-L treatment, which eventually led to a decrease in the F/B ratio in the obese mice.

In this study, different doses of GSDF increased the relative abundance of lactic acid bacteria and *Bacteroidetes*, decreased the relative abundance of *Dubosiella*, and affected the composition of the intestinal flora. *Bacteroidetes* and lactic acid bacteria are important components of the intestinal microbial community. *Bacteroidetes* increase intestinal permeability and accelerate lipolysis [47], whereas lactic acid bacteria inhibit the propagation and growth of harmful bacteria, such as *Staphylococcus aureus*, thereby maintaining the balance of intestinal flora and increasing body metabolism [48]. Overall, the outcomes of this study demonstrate that GSDF intervention gave rise to structural modifications in the intestinal flora of mice. It also resulted in an enhancement of growth performance and a boost in energy absorption. Whether the effect of GSDF is related to the components that are hydrolyzed by GSDF during the digestion process remains to be investigated. The relative abundance of *Dubosiella*, *Helicobacter*, and *Mycoplasma* in the GSDF treatment groups decreased significantly. In addition, according to the genus-level species abundance histogram, the relative abundances of *Faecalibaculum*, *Colidextribacter*, *Lachnospiraceae_Nk4A136_group*, *Lactobacillus*, and *Alloprevotella* increased after GSDF intervention.

To more clearly define the significance of the abundance difference between bacterial groups, we used LEfSe, which showed that beneficial bacteria such as *Bacteroides* had a significant impact on the control group. The main bacteria in the HFD group were harmful bacteria, such as *Dubosiella*. An increase in *Dubosiella* can lead to an imbalance in intestinal flora, abnormal energy metabolism, and exacerbation of obesity [49]. In addition, the dominant species in the GSDF intervention group were core intestinal flora such as *Trichospirillum* and *Desulfovibrio*. *Trichospirochetes* can produce long-chain fatty acids such as methyl oleate and palmitate. These fatty acids may aggravate obesity and insulin resistance induced by a high-fat diet [3], whereas *Desulfovibrio* can produce hydrogen sulfide, a metabolite with an inhibitory effect on gut hormone GLP-1, in turn disrupting the metabolic balance of the host to promote obesity [50].

## 5. Conclusions

In this study, we investigated the effects of GSDF on the PPAR/AMPK signaling pathway and intestinal health in obese mice. GSDF reduces body weight and fasting blood glucose index; improves liver, kidney, BAT, mWAT, and eWAT indexes; and restores the balance of TG, TC, LDL-C, and HDL-C in vivo. It also increases the content of serum cytokine IL-4, reduces the content of IL-1β, increases the expression level of PPAR-γ, AMPK, and P-AMPK protein, inhibits lipogenesis, improves lipid metabolism disorder, reduces the expression of IL-1β, TNF-α, and other proteins in epididymal adipose tissue, and improves tissue inflammation caused by obesity through regulating the PPAR/AMPK signaling pathway. By reducing the F/B ratio and adjusting the richness of *Firmicutes* and *Bacteroidetes*, cecal health was improved through altering the composition of lactic acid bacteria, including *Lactobacillus*, *Blautia*, *Lachnospiraceae_NK4A136_group*, *Alloprevotella*, and *Faecalibaculum*. These results suggest that GSDF may be an effective lipid-lowering agent that can ameliorate hyperlipidemia in obese mice through the modulation of the intestinal microbiota.

## Figures and Tables

**Figure 1 foods-14-01716-f001:**
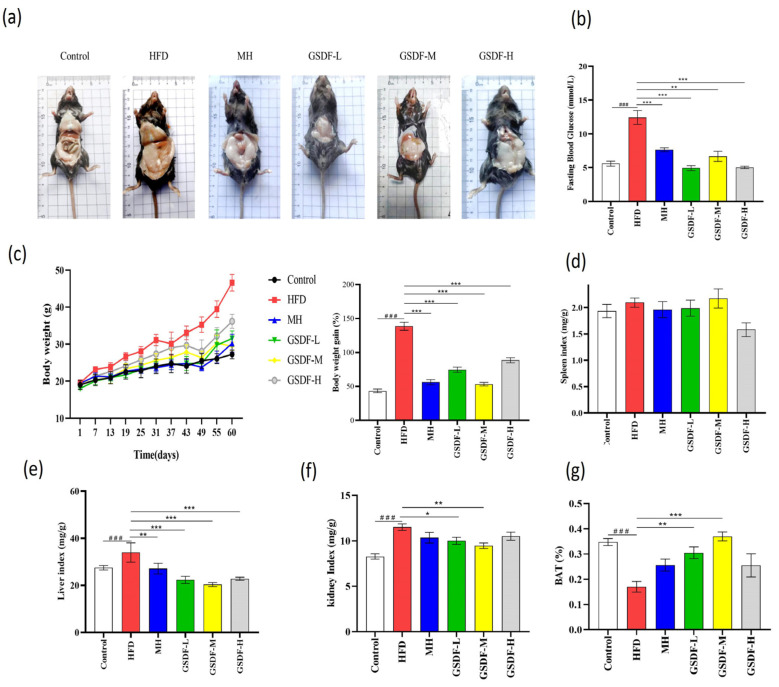
Effect of GSDF on basic indicators in HFD-induced obese mice. (**a**) Effects of GSDF on morphology in mice; (**b**) fasting blood glucose; (**c**) body weight and rate of body weight gain; (**d**) spleen index; (**e**) liver index; (**f**) kidney index; (**g**) BAT. Statistical significance was determined by a one-way analysis of variance (ANOVA). HFD, high-fat-diet group; MH, metformin hydrochloride group; GSDF-L, ginseng soluble dietary fiber low-dose group; GSDF-M, ginseng soluble dietary fiber medium-dose group; GSDF-H, ginseng soluble dietary fiber high-dose group; (*n* = 8, means ± SEM); ### *p* < 0.001 vs. control group, * *p* < 0.05 vs. HFD group, ** *p* < 0.01 vs. HFD group, *** *p* < 0.001 vs. HFD group.

**Figure 2 foods-14-01716-f002:**
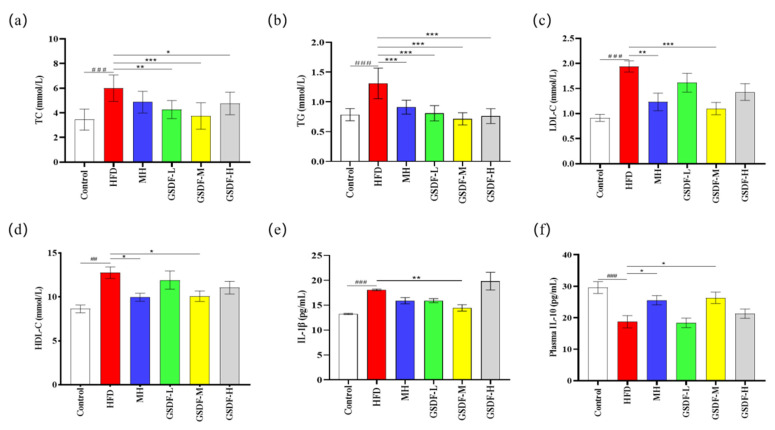
Effect of GSDF on serum cytokine and lipid in HFD-induced obese mice. (**a**) TC, (**b**) TG, (**c**) LDL-C, (**d**) HDL-C, (**e**) IL-1β, (**f**) IL-10. Statistical significance was determined by a one-way analysis of variance (ANOVA). HFD, high-fat-diet group; MH, metformin hydrochloride group; GSDF-L, ginseng soluble dietary fiber low-dose group; GSDF-M, ginseng soluble dietary fiber medium-dose group; GSDF-H, ginseng soluble dietary fiber high-dose group; (*n* = 8, means ± SEM); ## *p* < 0.01 vs. control group, ### *p* < 0.001 vs. control group, * *p* < 0.05 vs. HFD group, ** *p* < 0.01 vs. HFD group, *** *p* < 0.001 vs. HFD group.

**Figure 3 foods-14-01716-f003:**
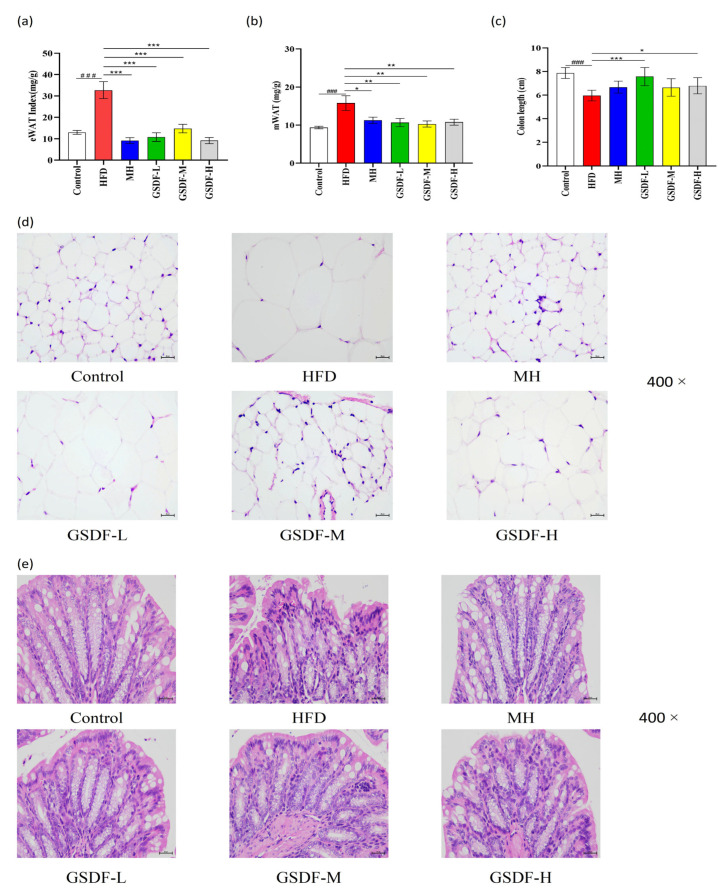
Effect of GSDF on the regulation of WAT and testicular histopathology in HFD-induced obese mice. (**a**) eWAT index, (**b**), mWAT index, (**c**) colon length, (**d**) histopathology of eWAT tissue (scale bar = 25 μm and objective: 400×), (**e**) mice colon histopathology (scale bar = 25 μm and objective: 400×). Statistical significance was determined by a one-way analysis of variance (ANOVA). HFD, high-fat-diet group; MH, metformin hydrochloride group; GSDF-L, ginseng soluble dietary fiber low-dose group; GSDF-M, ginseng soluble dietary fiber medium-dose group; GSDF-H, ginseng soluble dietary fiber high-dose group; (*n* = 8, means ± SEM); ### *p* < 0.001 vs. control group, * *p* < 0.05 vs. HFD group, ** *p* < 0.01 vs. HFD group, *** *p* < 0.001 vs. HFD group.

**Figure 4 foods-14-01716-f004:**
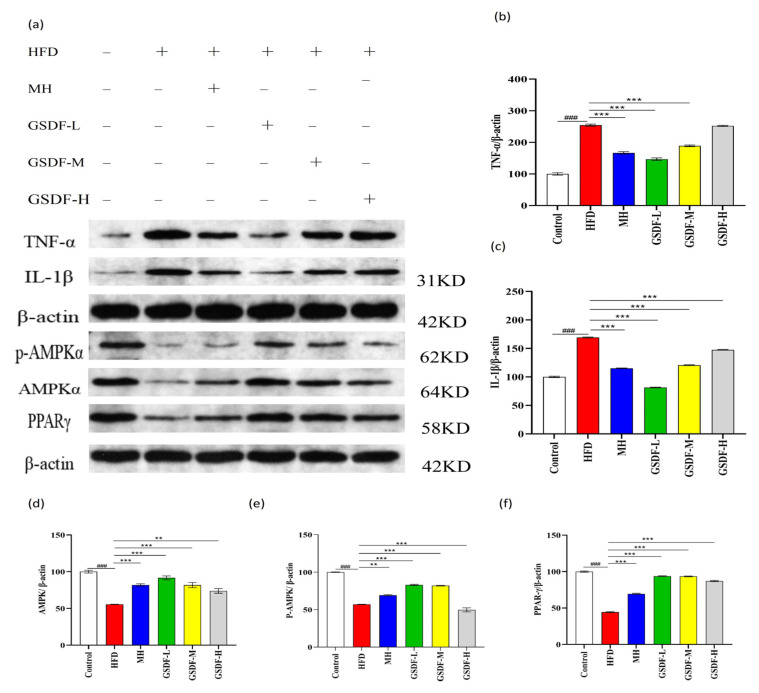
Effects of GSDF on AMPK pathway in mice. (**a**) Western blot images; (**b**) relative expression of TNF-α/β-actin; (**c**) relative expression of IL-1β/β-actin; (**d**) relative expression of AMPK/β-actin; (**e**) relative expression of PPAR-γ/β-actin; (**f**) relative expression of p-AMPK/β-actin. β-actin was used as the internal control. Statistical significance was determined by a one-way analysis of variance (ANOVA). HFD, high-fat-diet group; MH, metformin hydrochloride group; GSDF-L, ginseng soluble dietary fiber low-dose group; GSDF-M, ginseng soluble dietary fiber medium-dose group; GSDF-H, ginseng soluble dietary fiber high-dose group; (*n* = 8, mean ± SEM); ### *p* < 0.001 vs. control group, ** *p* < 0.01 vs. HFD group, *** *p* < 0.001 vs. HFD group.

**Figure 5 foods-14-01716-f005:**
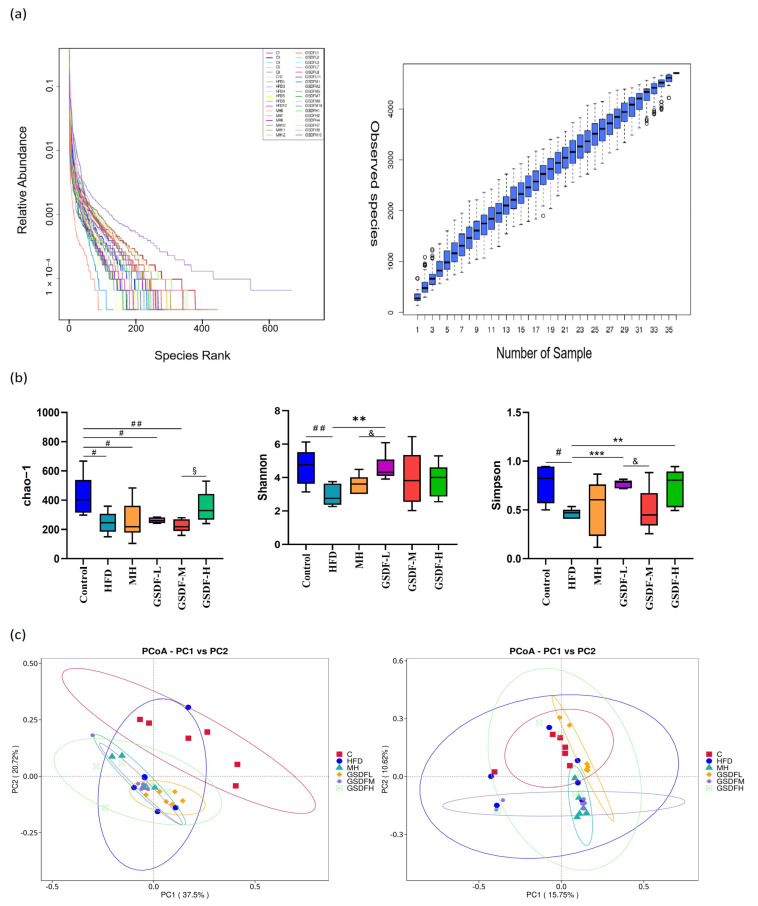
Effects of GSDF on the composition and diversity of intestinal flora in mice. (**a**) Alpha diversity analysis of intestinal flora in mice, (**b**), principal coordinate analysis (PCoA) showing the distances between different samples (Bray–Curtis distance, (**c**) statistical significance was determined by a one-way analysis of variance (ANOVA). HFD, high-fat-diet group; MH, metformin hydrochloride group; GSDF-L, ginseng soluble dietary fiber low-dose group; GSDF-M, ginseng soluble dietary fiber medium-dose group; GSDF-H, ginseng soluble dietary fiber high-dose group; (*n* = 6, mean ± SEM); # *p* < 0.05 vs. control group, ## *p* < 0.01 vs. control group, ** *p* < 0.01 vs. HFD group, *** *p* < 0.001 vs. HFD group, & *p* < 0.05 vs. GSDF-L group, § *p* < 0.05 vs. GSDF-H group.

**Figure 6 foods-14-01716-f006:**
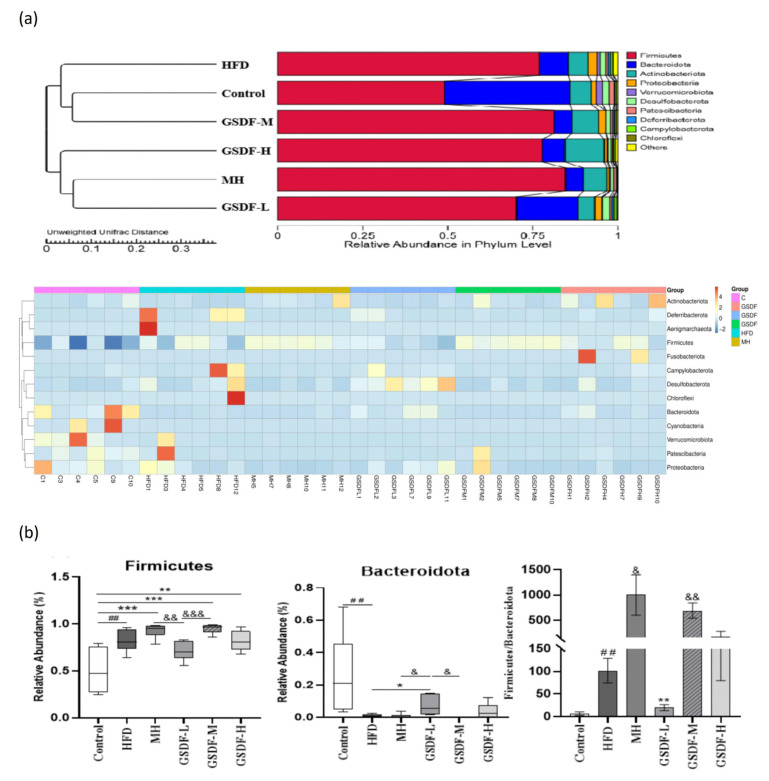
Effect of GSDF on the composition of intestinal flora at the portal level in mice. (**a**) Taxonomic composition at the phylum levels, (**b**) abundance comparison of major species at the phylum levels. Statistical significance was determined by a one-way analysis of variance (ANOVA). HFD, high-fat-diet group; MH, metformin hydrochloride group; GSDF-L, ginseng soluble dietary fiber low-dose group; GSDF-M, ginseng soluble dietary fiber medium-dose group; GSDF-H, ginseng soluble dietary fiber high-dose group; (*n* = 6, mean ± SEM); ## *p* < 0.01 vs. control group, * *p* < 0.05 vs. HFD group, ** *p* < 0.01 vs. HFD group, *** *p* < 0.001 vs. HFD group, & *p* < 0.05 vs. GSDF-L group, && *p* < 0.01 vs. GSDF-L group, &&& *p* < 0.001 vs. GSDF-L group.

**Figure 7 foods-14-01716-f007:**
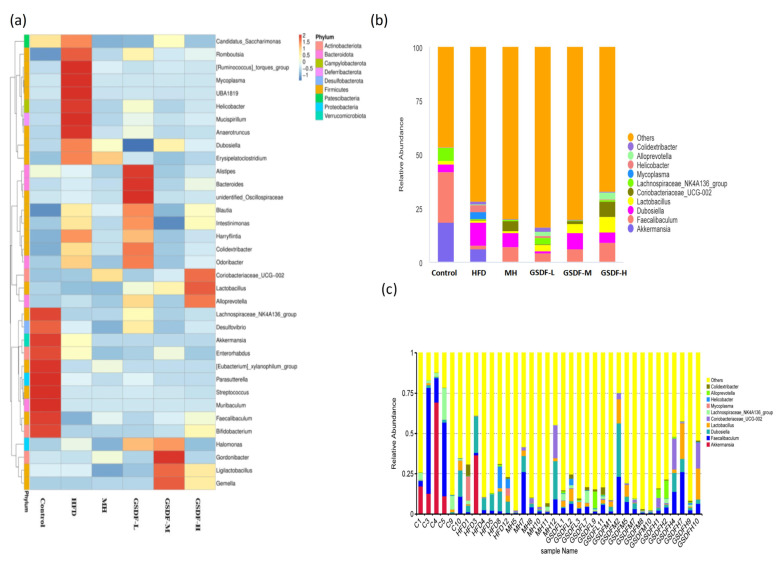
Effect of GSDF on the composition of intestinal flora at the genus level in mice. (**a**) Heat map of relative abundance at genus level. (**b**) Histogram of species abundance by group at the genus level. (**c**) Histogram of species abundance by sample at genus level. Statistical significance was determined by a one-way analysis of variance (ANOVA). HFD, high-fat-diet group; MH, metformin hydrochloride group; GSDF-L, ginseng soluble dietary fiber low-dose group; GSDF-M, ginseng soluble dietary fiber medium-dose group; GSDF-H, ginseng soluble dietary fiber high-dose group; (*n* = 6, mean ± SEM).

**Figure 8 foods-14-01716-f008:**
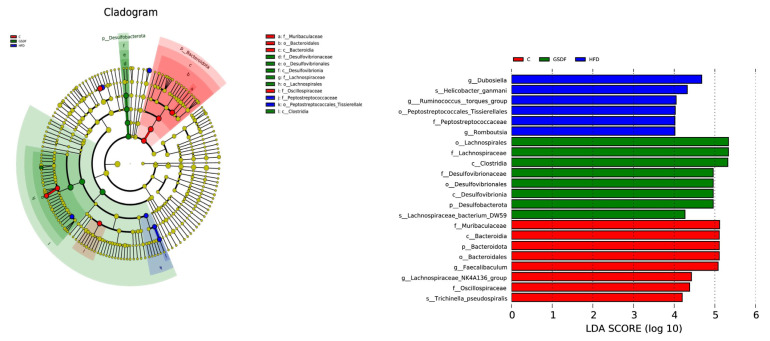
LEfSe analysis of GSDF on intestinal flora in mice. Statistical significance was determined by a one-way analysis of variance (ANOVA). HFD, high-fat-diet group; MH, metformin hydrochloride group; GSDF-L, ginseng soluble dietary fiber low-dose group; GSDF-M, ginseng soluble dietary fiber medium-dose group; GSDF-H, ginseng soluble dietary fiber high-dose group; (*n* = 6, mean ± SEM).

## Data Availability

The original contributions presented in this study are included in the article. Further inquiries can be directed to the corresponding author.

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
