# Peer review of "Ginseng Soluble Dietary Fiber Reverses Obesity via the PPAR/AMPK Signaling Pathway and Improves Intestinal Flora in Mice"

_foods, 2025, doi:10.3390/foods14101716_

Round 1

Reviewer 1 Report

Comments and Suggestions for Authors

In my opinion, the work presented by Zhang et al., is very interesting and addresses an issue that is still little known in the literature. The work is well written. However, there are some aspects that could be improved.

Minor revision:

  • Lane 159: specify the amount of protein samples separated by SDS-PAGE
  • Microbiome analysis and statistical analysis: The authors should explain in the materials and methods the bionformatics and statistical analysis used to study the microbiome. In particular, they should describe the type of indices used (Chao-1, Shannon and Simpson index); how they performed the PCA and LEfSe analysis.
  • The authors describe organ indices in the results (section 3.1). How were these indices derived? They should be described in the materials and methods.
  • Section 3.1: better specify the non-significant effect of the GSDF-H group
  • In some cases, the authors claim that the GSDF-M group had better results than the other groups but, in the graphs presented, comparisons are always made between Control vs. HDF and HDF vs. GSDF groups. In order to assert that one GSDF group has a better response than another, a statistical analysis comparing the GSDF groups must also be included.
  • Lane 319: Figure 5b shows that the shannon index of the GSDF-L group is higher than that of MH group.
  • Furthermore, while the divergent impact of the three GSDF groups on the analysed parameters is frequently highlighted in the results, this subject is not addressed in sufficient depth in the discussion. Do the authors have any hypotheses to explain the different behaviour of the three groups?
  • Lane 256: Fig. 3 c, e

Author Response

Comments 1: Lane 159: specify the amount of protein samples separated by SDS-PAGE.

Response 1: Thank you for pointing this out. We agree with this comment. Therefore, we have added it to page 4, line 172, and highlighted it in red.

Comments 2: Microbiome analysis and statistical analysis: The authors should explain in the materials and methods the bionformatics and statistical analysis used to study the microbiome. In particular, they should describe the type of indices used (Chao-1, Shannon and Simpson index); how they performed the PCA and LEfSe analysis.

Response 2: Thank you very much for the reviewer's suggestion, and we offer the following explanation in response to your query:

Alpha - Diversity: we calculated Chao-1, Shannon and Simpson indices. Chao-1 estimates species richness based on single and double instances. Shannon and Simpson indices consider both richness and evenness. Beta Diversity: PCA was performed to visualize the overall pattern of microbial community structure in the samples. Relative abundance data of OTUs in the samples were used as input for PCA, one of the β-diversity analyses. Lesfe analysis is the identification of microbial taxa that are rich in variation among different groups. It combines statistical significance tests with biological consistency checks. LDA was used to estimate the effect size of each differentially abundant feature. features with LDA scores > threshold and p-values < significance level were considered to have significant differential abundance.

They are all statistically significant and they are using different perspectives to analyze the variability in gut flora richness as well as microbial community structure.

Comments 3: The authors describe organ indices in the results (section 3.1). How were these indices derived? They should be described in the materials and methods.

Response3: Thank you very much for the reviewer's suggestion, we are very supportive of it. Therefore, we have added it to page 4, line 150, and marked it in red.

Comments 4: Section 3.1: better specify the non-significant effect of the GSDF-H group.

Response 4: The reviewer's comments are greatly appreciated and the errors in the manuscript were corrected by revising lines 211 and 216 on page 5 of the article and marked them in red.

Comments 5: In some cases, the authors claim that the GSDF-M group had better results than the other groups but, in the graphs presented, comparisons are always made between Control vs. HDF and HDF vs. GSDF groups. In order to assert that one GSDF group has a better response than another, a statistical analysis comparing the GSDF groups must also be included.

Response5: Thank you very much for the reviewer's suggestion, due to our negligence, we now have no more evidence that the medium-dose group has the best effect, and we will follow up with further research on the effect of different doses of GSDF and the mechanism of action. Therefore, we have revised the ambiguous statements in the article.

Comments 6: Lane 319: Figure 5b shows that the shannon index of the GSDF-L group is higher than that of MH group.

Response 6: Thanks to the reviewer's suggestion, a clerical error in line 339 on page 10 of the manuscript has been corrected and highlighted in red.

Comments 7: Furthermore, while the divergent impact of the three GSDF groups on the analysed parameters is frequently highlighted in the results, this subject is not addressed in sufficient depth in the discussion. Do the authors have any hypotheses to explain the different behaviour of the three groups?

Response 7: Thank you very much to the reviewers for their suggestions. We added this portion of the description in the discussion section on page 16, line 495, and highlighted it in red.

Comments 8: Lane 256: Fig. 3 c, e.

Response 8: Many thanks to the reviewers for their comments. Figure 3 c shows the length of the colon and Figure 3 e shows histopathological sections of the colon. These two experiments were done to echo the intestinal flora.

Reviewer 2 Report

Comments and Suggestions for Authors

The study explores the effects of Ginseng Soluble Dietary Fiber (GSDF) on obesity and intestinal flora in high-fat diet-induced obese mice. The research aims to elucidate the mechanisms of GSDF, focusing on the PPAR/AMPK signaling pathway and gut microbiota modulation.

Comments

- Weaknesses: Dose-dependency inconsistencies: The medium dose (GSDF-M) appears to be most effective in many cases, but reasons for the lack of dose-dependent trends are not discussed.

The experimental design used metformin as a positive control group to assess the effects of GSDF in mice with high-fat diet-induced obesity. However, metformin is a drug approved primarily for the treatment of type 2 diabetes, not specifically for obesity. Could you provide a clearer rationale for the choice of metformin as a comparator in this model? It would be useful to know whether its selection was based on previous evidence of its efficacy in modulating lipid metabolism or gut microbiota in obesogenic models, and whether there are references supporting its use as a positive control in studies focused exclusively on obesity.

Author Response

Comments 1: Weaknesses: Dose-dependency inconsistencies: The medium dose (GSDF-M) appears to be most effective in many cases, but reasons for the lack of dose-dependent trends are not discussed.

Response 1: Many thanks to the reviewers for their comments. As we explored the dose-dependence, it is highly likely that the differential performance presented by the three different dose GSDF groups is closely linked to their respective structural properties. As the key constituents of GSDF, structural elements such as the molecular weight size, the proportion of various types of monosaccharides, the specific type of glycosidic bonds, and the overall spatial conformation of the polysaccharides all play a role in their biological activity. In view of this, the polysaccharides in the three different doses of GSDF groups might have significant differences in the above structural features, which is the reason why different doses of GSDF produced different effects in improving obesity and regulating intestinal flora. Therefore, we have modified our conclusions containing differences. We added this portion of the description in the discussion section on page 16, line 495, and highlighted it in red.

Comments 2: The experimental design used metformin as a positive control group to assess the effects of GSDF in mice with high-fat diet-induced obesity. However, metformin is a drug approved primarily for the treatment of type 2 diabetes, not specifically for obesity. Could you provide a clearer rationale for the choice of metformin as a comparator in this model? It would be useful to know whether its selection was based on previous evidence of its efficacy in modulating lipid metabolism or gut microbiota in obesogenic models, and whether there are references supporting its use as a positive control in studies focused exclusively on obesity.

Response 2: Thank you very much for your detailed and specialized queries on the selection of the positive control drug in our experimental design, which prompted us to review and elaborate more deeply on the rationale for choosing metformin as a control.

We chose metformin as a positive control not only because of its widespread clinical use, but more importantly based on prior research evidence of its efficacy in modulating lipid metabolism and gut microbiota in obesogenic models. This evidence provides a solid theoretical basis for our choice of metformin, and there are also relevant references supporting its use as a positive control in obesity-related studies. The references are listed below.

Sun L, Xie C, Wang G, et al. Gut microbiota and intestinal FXR mediate the clinical benefits of metformin. Nat Med. 2018;24(12):1919-1929.
Dong J, Tong X, Xu J, el at. Metformin improves obesity-related oligoasthenospermia via regulating the expression of HSL in testis in mice. Eur J Pharmacol. 2024, 968: 176388.

Reviewer 3 Report

Comments and Suggestions for Authors

The paper is overall well-written and has a clear and logical flow. As for methodology covering analysis evaluating physiological indices, levels of blood lipids and serum cytokine (TG, TC, LDL-C, and HDL-C, and IL-1β, IL-10), and the protein expression levels of the AMPK/PPAR signaling pathway, I have no objection. As for analysis based on the 16S rRNA detection of intestinal microbiome, as a measure reflecting cecum health, could the authors describe it in more detail? Also, primer sequences are missing. The explanation describing how the structure of the intestinal microbiota was examined at the taxonomic levels, as well. Please, provide more details.

The paper Ref. No. Foods-3601579 I have understood as a continuation of the authors’ previous work (referred under the reference 25. Hua, M.; Sun, Y.; Shao, Z.; Lu, J.; Lu, Y.; Liu, Z. Functional soluble dietary fiber from ginseng residue: Polysaccharide characterization, structure, antioxidant, and enzyme inhibitory activity. J. Food Biochem. 2020, 44, e13524.), which results were described by few lines (71-79) in the introduction part.

Perhaps the authors should elaborate in more details results related to the GSDF composition, because is important for understanding the activity. When mentioning this, would be interesting to consider how additional experiment mimicking digestion conditions would reflect on the GSDF compositional change since the food containing different doses of GSDF upon digestion is certainly influenced the activity of the experiment. Antidiabetic regulatory effects of ginsenosides, polysaccharides, and polyacetylenic alcohols are well known; could the effect of GSDF feed be related some of components originating from hydrolyzed GSDF? In my opinion, the additional experiment/discussion would greatly contribute to the quality of the research.

The structural balance of the intestinal flora is important and the authors experimentally shown the prebiotic potential of GSDF by the modulation of gut microbiota however, it is important to understand whether compounds present in the GSDF complex matrices can act as AMPK agonists. As the authors in their previously published paper (Hua, M.; Sun, Y.; Shao, Z.; Lu, J.; Lu, Y.; Liu, Z. Functional soluble dietary fiber from ginseng residue: Polysaccharide characterization, structure, antioxidant, and enzyme inhibitory activity. J. Food Biochem. 2020, 44, e13524) stated “A large number of studies have shown that the antioxidant activity of dietary fiber is related to its polysaccharide structure and small molecule active substances, such as ginsenosides and polyphenols.” In sense of AMPK activation, can you apply similar discussion to the current research?

I have also observed that the attached file containing original images are missing tags. For clarity, could the authors please add them?   For the above reason my suggestion would be a major revision.  

Author Response

Comments 1: The paper is overall well-written and has a clear and logical flow. As for methodology covering analysis evaluating physiological indices, levels of blood lipids and serum cytokine (TG, TC, LDL-C, and HDL-C, and IL-1β, IL-10), and the protein expression levels of the AMPK/PPAR signaling pathway, I have no objection. As for analysis based on the 16S rRNA detection of intestinal microbiome, as a measure reflecting cecum health, could the authors describe it in more detail? Also, primer sequences are missing. The explanation describing how the structure of the intestinal microbiota was examined at the taxonomic levels, as well. Please, provide more details.

Response 1: (1) Thank you very much for the reviewer's comments, which we strongly support. 16S rRNA gene is characterized by sequence differences in the variable region, which can be used as a bacterial taxonomic marker, and the gene is commonly found in bacteria, not eukaryotes, which makes it a broad and specific test. 16S rRNA analysis can be used to analyze the composition of microbial communities and the relative abundance of different bacteria, and it can also be used for diversity analysis, with high α-diversity in healthy cecum microbial communities and differences in β-diversity among cecum samples with different health conditions. 16S rRNA analysis can analyze the composition of microbial communities and the relative abundance of different bacteria through DNA extraction, PCR amplification, sequencing and database comparison. The α-diversity of healthy cecum microbial communities is high and the β-diversity varies among cecum samples of different health conditions.

(2) Thank you very much for the reviewer's comments, our primer information is:

Upstream primer: F CCTAYGGGRBGCASCAG

Downstream primer: R GGACTACNNGGGGTATCTAAT

We spliced and filtered the raw data obtained from sequencing (Raw Data) to obtain the valid data (Clean Data), and then obtained the final ASVs by noise reduction based on the valid data through the method of DADA2.The classify-sklearn algorithm of QIIME2 was used to annotate the species for each ASV using the pre-trained Naive Bayes classifier for species annotation. The annotation database was Silva 138.1, and species abundance tables at the level of kingdom, phylum, class, order, family, genus and species were obtained based on the results of the annotation of ASVs and the characterization table of each sample.It was used to examine the structure of the gut microbiota.

Comments 2: The paper Ref. No. Foods-3601579 I have understood as a continuation of the authors’ previous work (referred under the reference 25. Hua, M.; Sun, Y.; Shao, Z.; Lu, J.; Lu, Y.; Liu, Z. Functional soluble dietary fiber from ginseng residue: Polysaccharide characterization, structure, antioxidant, and enzyme inhibitory activity. J. Food Biochem. 2020, 44, e13524.), which results were described by few lines (71-79) in the introduction part.

Response 2:Thank you very much to the reviewers for their comments, we are very supportive. Therefore, we have added this on page 2, line 85, and highlighted it in red.

Comments 3: Perhaps the authors should elaborate in more details results related to the GSDF composition, because is important for understanding the activity. When mentioning this, would be interesting to consider how additional experiment mimicking digestion conditions would reflect on the GSDF compositional change since the food containing different doses of GSDF upon digestion is certainly influenced the activity of the experiment. Antidiabetic regulatory effects of ginsenosides, polysaccharides, and polyacetylenic alcohols are well known; could the effect of GSDF feed be related some of components originating from hydrolyzed GSDF? In my opinion, the additional experiment/discussion would greatly contribute to the quality of the research.

Response 3: Many thanks to the reviewers for their comments, which we very much support. Therefore, We added this element to the discussion and marked it in red on page 17, line 537. Whether the effect of GSDF is related to the composition of the hydrolyzed GSDF will be followed up with experimental studies by other members of our lab.

Comments 4: The structural balance of the intestinal flora is important and the authors experimentally shown the prebiotic potential of GSDF by the modulation of gut microbiota however, it is important to understand whether compounds present in the GSDF complex matrices can act as AMPK agonists. As the authors in their previously published paper (Hua, M.; Sun, Y.; Shao, Z.; Lu, J.; Lu, Y.; Liu, Z. Functional soluble dietary fiber from ginseng residue: Polysaccharide characterization, structure, antioxidant, and enzyme inhibitory activity. J. Food Biochem. 2020, 44, e13524) stated “A large number of studies have shown that the antioxidant activity of dietary fiber is related to its polysaccharide structure and small molecule active substances, such as ginsenosides and polyphenols.” In sense of AMPK activation, can you apply similar discussion to the current research?

Response 4: Many thanks to the reviewers for their comments, which we are very supportive of. Therefore, we have added this discussion and marked it in red on page 15, line 464.

Comments 5: I have also observed that the attached file containing original images are missing tags. For clarity, could the authors please add them? For the above reason my suggestion would be a major revision.

Response 5: Many thanks to the reviewers for their comments. We have reviewed the original images and resent the collated original images to the editors. The reviewers are invited to consult them.

Round 2

Reviewer 3 Report

Comments and Suggestions for Authors

The authors have provided the necessary updates for which I have requested, the paper could be considered acceptable for publication.

Sincere greetings